# Compressive three-dimensional super-resolution microscopy with speckle-saturated fluorescence excitation

M. Pascucci[1], S. Ganesan[1], A. Tripathi[2,3], O. Katz[2], V. Emiliani [1] & M. Guillon[1]

Nonlinear structured illumination microscopy (nSIM) is an effective approach for super-resolution wide-field fluorescence microscopy with a theoretically unlimited resolution. In nSIM, carefully designed, highly-contrasted illumination patterns are combined with the saturation of an optical transition to enable sub-diffraction imaging. While the technique proved useful for two-dimensional imaging, extending it to three-dimensions is challenging due to the fading of organic fluorophores under intense cycling conditions. Here, we present a compressed sensing approach that allows 3D sub-diffraction nSIM of cultured cells by saturating fluorescence excitation. Exploiting the natural orthogonality of speckles at different axial planes, 3D probing of the sample is achieved by a single two-dimensional scan. Fluorescence contrast under saturated excitation is ensured by the inherent high density of intensity minima associated with optical vortices in polarized speckle patterns. Compressed speckle microscopy is thus a simple approach that enables 3D super-resolved nSIM imaging with potentially considerably reduced acquisition time and photobleaching.

[1] Neurophotonics Laboratory UMR8250, University Paris Descartes, 47 rue des Saints-Pères, 75270 Paris, France. [2] Department of Applied Physics, The Hebrew University of Jerusalem, Jerusalem 9190401, Israel. [3] Department of Physics, Indian Institute of Technology, Delhi 110016, India. Correspondence and requests for materials should be addressed to M.G. (email: marc.guillon@parisdescartes.fr)

Super-resolution fluorescence microscopy has proven to be a fundamental tool in biology to unveil processes occurring at the nano-scale[1]. Subdiffraction imaging can be obtained by exciting the fluorescent probes in a nonlinear regime, either inducing binary stochastic responses[2,3] or under saturated conditions[4,5]. The saturated optical transition can be absorption[5,6], stimulated emission[4,7], ground state depletion[8] or fluorescence photo-switching, generalized as REversible Saturable Optical Linear Fluorescence Transitions (RESOLFT) techniques[9,10]. Illumination with patterned light excitation then appears as an essential scheme[11,12].

In the linear excitation regime, structured illumination microscopy (SIM) with fringes[13], grids[14], or speckles[15–17] can provide up to a twofold improvement in resolution as compared to widefield imaging, especially in three dimensions (3D)[18]. Imaging in 3D is then obtained by acquiring every single transverse plane. In this configuration, fluorescence arising from out-of-focus planes is thus induced to be wasted since suppressed by numerical sectioning[19]. Fluorescence-signal-wasting is all the more detrimental under optical-saturation conditions, required to allow breaking the diffraction limit up to theoretically unlimited resolutions, in which case nonlinear photo-bleaching may occur[5,20]. Typically, the fragility of dyes has never allowed recording several transverse planes in saturated-excitation structured-illumination microscopy. 3D fluorescence nanoscopy requirements thus call for compressed sensing approaches[21].

Random wavefields feature two main interesting properties that make them suitable for 3D super-resolution microscopy. First, speckle patterns lying in different transverse planes are orthogonal relatively to the cross-correlation product (Supplementary Note 2), thus allowing axial discrimination of two-dimensional objects[22,23]. This property exactly provides the random-projection-measurement configuration ideally suited for compressed imaging reconstruction[24,25], in particular for 3D imaging[26]. Second, speckles exhibit strong intensity contrasts since naturally containing a high density of optical vortices of topological charge one[27], such as typically used in STimulated Emission Depletion (STED) microscopy[28]. Optical vortices in speckles are associated with nodal lines of intensity (in 3D) that can confine fluorescence to subdiffraction dimensions[29], despite the contribution of the axial field. Indeed, in tightly focused beams, the vectorial nature of light cannot be neglected, especially under saturated conditions, in which case the axial field may destroy most of intensity zeros. Whether it is possible to break the diffraction barrier with saturated speckle patterns has remained an open theoretical question[30].

Here, we demonstrate the possibility to achieve 3D super-resolution imaging by a single 2D raster scan under saturated fluorescence excitation with tightly focused speckle patterns. Using a custom-built speckle scanning microscope, we can image with a factor 3.3 beyond the diffraction barrier. Super-resolution imaging capabilities are characterized using fluorescent nano-beads, and applied for imaging stained lysosomes in fixed cultured cells.

## Results

**Experimental setup**. The scanning speckle microscope is sketched in Fig. 1a. A spatial light modulator (SLM), conjugated to the back focal plane of the objective lens, is used to generate a 3D fully developed speckle pattern (See Methods and Supplementary Fig. 1 for a more complete description). A regular diffuser could replace the SLM but the latter allows a dynamic control of the size of the illuminating speckle pattern and thus of the intensity at the sample plane. The random wave entering the objective is circularly polarized in order to minimize the axial field at isotropic

vortices of same handedness and so, to provide isotropic super-resolution in transverse planes under saturated excitation conditions (see Supplementary Note 8).

**Optical saturation**. To quantify the saturation excitation level, we may use a two-energy-level-dye model[31]. The excitation probability of the dye typically depends on several parameters such as the fluorescence lifetime of the dye $\tau_f$, its absorption cross-section $\sigma$, and the laser-pulse temporal intensity profile (width $\tau_p$, amplitude $I_p$ and shape). A subnanosecond laser is used, delivering pulses shorter than the fluorescence life-time of the dye ($\tau_p \sim 500$ ps) in order to efficiently saturate the optical transition with the minimal average power, and long enough for keeping a low-multiphoton absorption probability. The repetition rate of 4 kHz is low enough to ensure dark-state relaxation between excitation pulses and so, to minimize photo-bleaching via inter-system crossing[20]. When exciting fluorescence with pulses much shorter than the fluorescence lifetime ($\tau_p \ll \tau_f$), the fluorescence signal may be approximated by $F(s) \simeq 1 - e^{-s}$ (See Supplementary Note 6), where we define $s$ as the saturation parameter. In the case of a step-wise pulse, $s = \frac{\sigma I_p \tau_p}{h\nu}$ where $h\nu$ is the quantum of excitation-light energy. For characterizing excitation saturation in our experiment, we illuminated a thin layer of beads with a speckle pattern and collected the average fluorescence signal. When averaging over intensity fluctuations of a fully developed speckle pattern (with probability density function $\rho(I) = 1/\langle I\rangle \exp(-I/\langle I\rangle)$), the average fluorescence signal can be derived analytically as (Supplementary Note 7):

$$\langle F \rangle = \frac{\langle s \rangle}{\langle s \rangle + 1}, \qquad (1)$$

where the notation $\langle \cdot \rangle$ stands for spatial averaging. The experimental fluorescence curve shown in Fig. 1b is thus fitted with this function. More conveniently, the average saturation parameter $\langle s \rangle$ can be expressed as $\langle s \rangle = \epsilon/\epsilon_s$, with $\epsilon = I_p \tau_p A$ the pulse energy ($A$ being the speckle spot surface) and $\epsilon_s = h\nu A/\sigma$ the pulse excitation energy for which fluorescence reaches half the maximum signal. In Fig. 1b, we measured $\epsilon_s = 640$ pJ for a 10 µm speckle spot.

**3D-point spread function characterization**. The intensity profile of a 3D speckle pattern (Fig. 1c) in a transverse $xy$-plane (Fig. 1d, e) and in an axial $xz$-plane (Fig. 1f, g) is shown both under nonsaturated (Fig. 1d, f) and saturated (Fig. 1e, g) excitation conditions. Speckle point spread functions (SPSF) were obtained by scanning isolated fluorescent nano-beads. Under saturated illumination conditions, dark round-shaped points remain both in the $xy$ and in the $xz$ sections (highlighted by dashed ellipses in Fig. 1e, g), which can be attributed to the crossing of these planes by nodal vortex lines. These dark points ensure contrast conservation under saturated excitation, so enlarging the power spectrum of the SPSF (Fig. 1h, i) and demonstrating the larger accessible spectral support of the optical transfer function for imaging applications. Contrast conservation under saturated conditions is at the basis of RESOLFT microscopy, in which resolution typically scales as[32]:

$$\delta x = \frac{\lambda}{2\text{NA}} \frac{1}{\sqrt{1+s}}. \qquad (2)$$

Although the average saturation level in Fig. 1e, g looks modest ($\langle s \rangle = 3.7$) as compared with typical saturation levels used in RESOLFT microscopy, this value is averaged over intensity fluctuations of the speckle, meaning that locally the saturation can be considerably higher. Importantly, the field gradient, can be large at the vortex centers where the field is minimum, ensuring

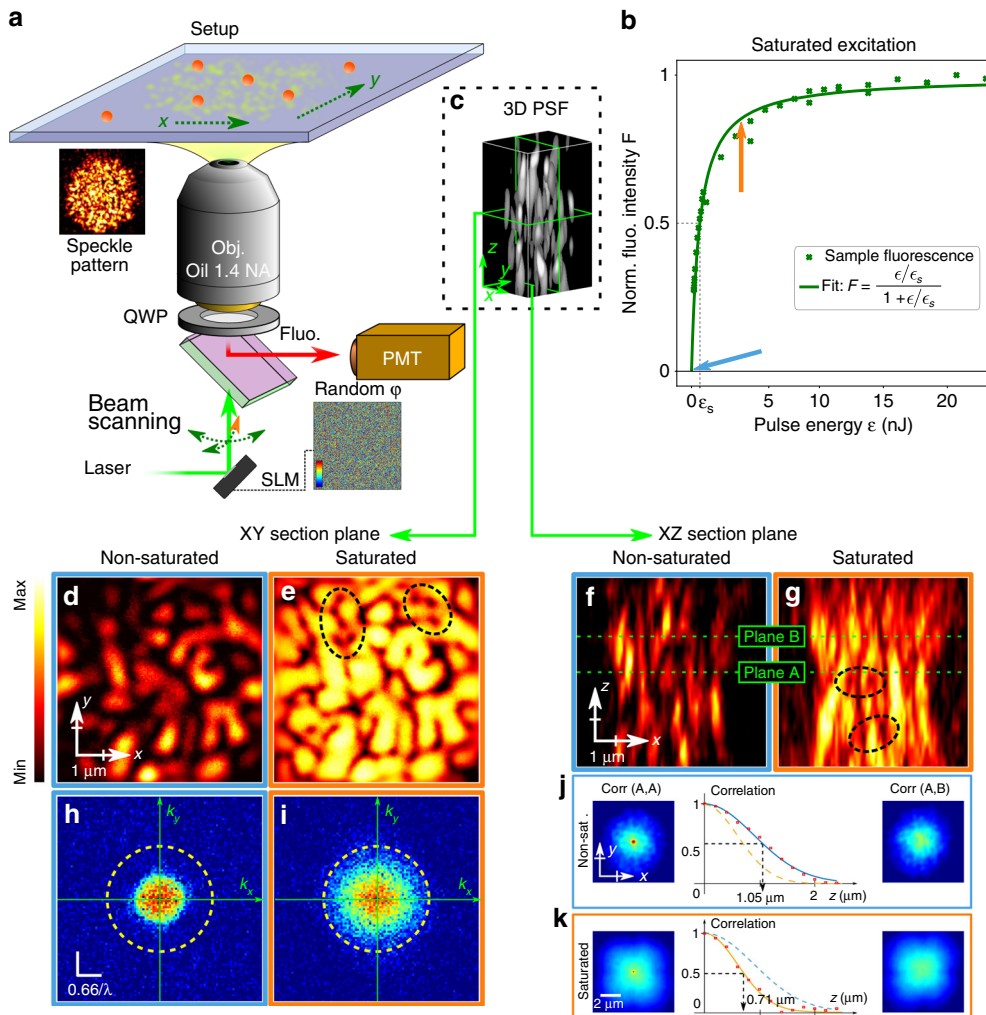

**Fig. 1** Principle of our speckle scanning microscope. A circularly polarized random wave-field generated by a spatial light modulator (SLM)—displaying a random phase mask—and a quarter wave-plate (QWP) is focused and scanned through an objective lens onto a fluorescent sample (**a**). The fluorescence signal from a thin and dense layer of fluorescent beads illuminated with a speckle is plotted in (**b**), as a function of the exciting-pulse energy. The curve is fitted with Eq. (1). The 3D speckle point spread function (SPSF) can be experimentally characterized by scanning a single fluorescent nano-bead in three dimensions (**c**). The speckle in the transverse plane (**d**, **e**) and in a longitudinal plane (**f**, **g**) are shown for low pulse-energy ($\langle s \rangle = 5 \times 10^{-3}$ in (**d**, **f**) and for high pulse-energy ($\langle s \rangle = 3.7$) in (**e**, **g**). Black dotted ellipses in (**e**) and (**g**) point out dark points identifying the plane crossing by optical vortex lines. The power spectra of SPSFs in (**d**, **e**) are represented in (**h**, **i**), respectively, illustrating the transverse power spectrum enlargement due to saturated excitation (dotted circles in (**h**, **i**) materialize the spectrum boundary in (**i**), as a visual reference). Cross-correlation products between different transverse planes A and B is shown in (**j**, **k**), in the nonsaturated and in the saturated case, respectively. In both regimes, the plots show that the cross-correlation peak vanishes with increasing defocus under saturated excitation conditions. Saturation decreases the correlation width

efficient spatial spectrum broadening under saturated conditions. Breaking the diffraction limit can thus be achieved by saturating an optical transition with speckles. However, super-resolution imaging implies exciting fluorescence up to the probes exhaustion, which is hardly compatible with 3D imaging. In this regard, imaging with random speckles is especially suited, thanks to their orthogonality properties.

2D speckle patterns appearing in different transverse planes are statistically orthogonal relatively to the cross-correlation product. When 2D-scanning a 3D sample with a speckle, each plane of the object is convolved with a distinct pattern. The resulting image is thus the linear combination of the contribution of all the planes. Since speckles are orthogonal, each plane can then be retrieved either by projection[23] or compressive sensing schemes[26]. To be precise, speckles are only orthogonal for large-enough axial separations: $\delta z \geq 2n\lambda/\mathrm{NA}^2$. When saturating an optical transition, not only saturated speckles remain orthogonal but this minimal

separation distance is even reduced. The cross-correlation of two $z$-distant transverse planes A and B is illustrated in Fig. 1j, k for the nonsaturated and the saturated case, respectively, and for $z = 0$ ($A = B$) and $z \gg 2n\lambda/\mathrm{NA}^2$. The plot of the cross-correlation product as a function of $z$ shows that the correlation distance is shorter under saturated conditions by a factor $\simeq \sqrt{2}$. Saturating the speckle patterns thus increases the axial density of modes, so-demonstrating the possibility to improve resolution along the propagation axis by optical saturation.

**3D super-resolution demonstration.** Imaging with random structures raises the specific challenge of object reconstruction. Techniques have been developed to reconstruct images even when the speckles are unknown[15,17] especially through scattering samples[33–36]. Adding sparsity constraints to the rebuilt object is particularly helpful, especially to retrieve 3D information from

2D images[26]. Adding sparsity priors about the object in compressive sensing approaches may even provide details smaller than the resolution of the instrument[36]. Conversely, in our case, super-resolution information is experimentally extracted from the sample thanks to the power spectrum enlargement of the optical transfer function under saturated fluorescence excitation. In order to demonstrate so, we first perform plane-by-plane Wiener deconvolution[37,38]. Interestingly, Wiener deconvolution being just a cross-correlation product with a spectral renormalization, speckles satisfy the same orthogonality properties (see

Supplementary Note 3). A single parameter must be adjusted: the mean power spectral density of the noise of images. In our results, we tuned this parameter by visual image optimization.

2D-scanned images of a sample consisting in three fluorescent beads located at different axial positions, are shown in Fig. 2a, b, under nonsaturated ($\langle s \rangle \ll 1$) and saturated conditions ($\langle s \rangle = 1.4$), respectively. The experimental characterization of the 3D-SPSFs, required for reconstructing the object, is achieved by 3D-scanning of single isolated beads with the speckle (Fig. 2c) both in the linear and in the saturated regime. Plane-by-plane Wiener

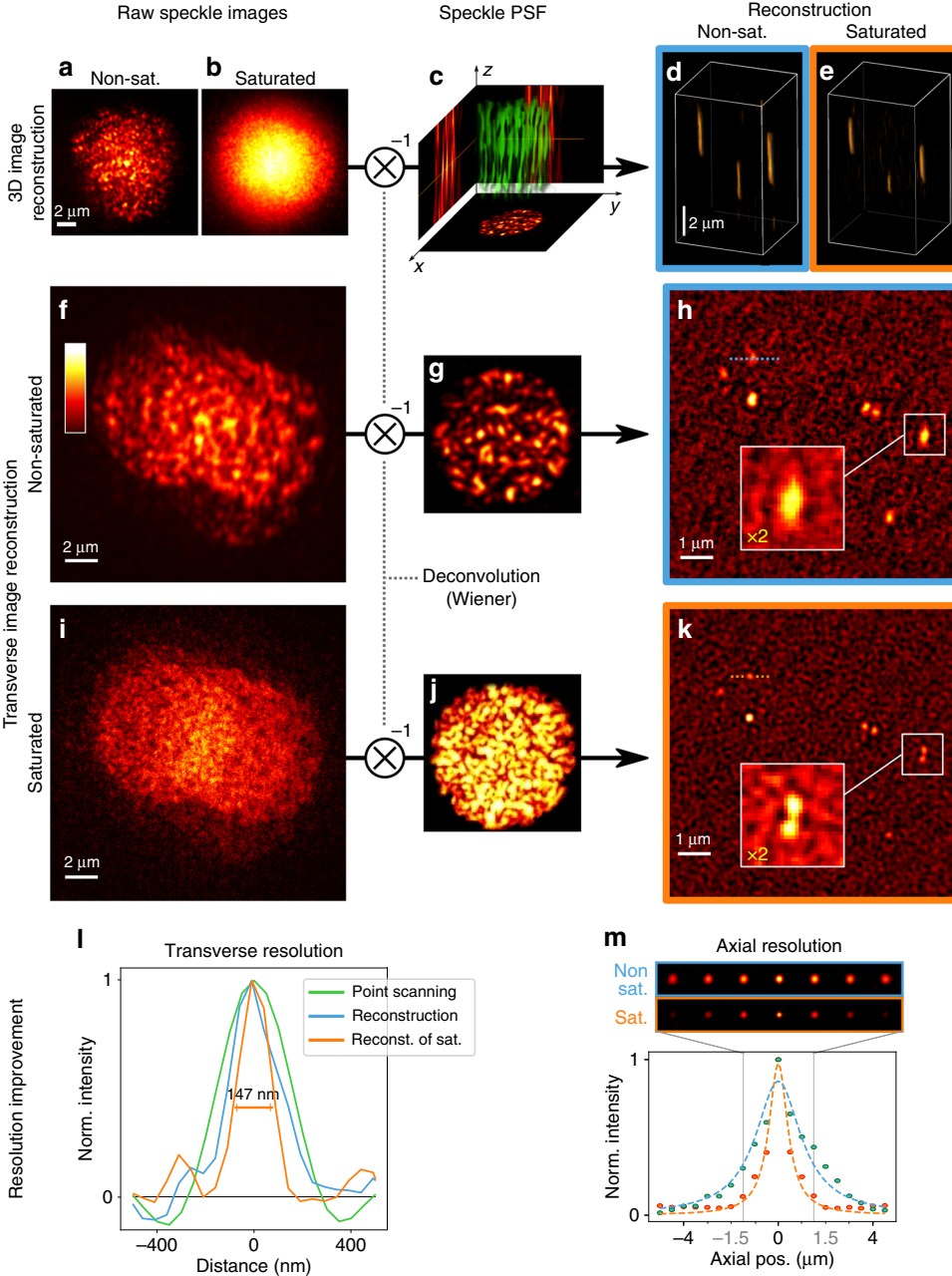

**Fig. 2** 3D super-resolution capabilities. 2D-scanning of fluorescent nano-beads by speckles (**a**, **b**) allows 3D object reconstruction (**d**, **e**). Image reconstruction is achieved by plane-by-plane Wiener deconvolution thanks to the prior experimental characterization of the 3D-SPSF (**c**). Speckle images were taken under non-saturated (**a**, **f**) and saturated (**b**, **i**) conditions. By depositing fluorescent 100 nm-beads on a coverslip, a bead cluster could be observed to be only resolved under saturated conditions (**k**) and not by linear-excitation speckle imaging (**h**). Line profiles in (**h**, **k**) are plotted in (**l**) and compared to the profile obtained by deconvolved point-scanning imaging (Supplementary Fig. 12). Resolution is improved in the speckle imaging mode as compared to point-scanning mode and super-resolution is obtained under saturated excitation conditions. Axial resolution improvement is shown in **m** where an axial bead intensity profile is plotted both in the linear and in the saturated excitation regimes. In all images, NA = 0.77 and saturated images were recorded with an average saturation parameter $\langle s \rangle = 1.4$

deconvolution of a 2D speckle image yields its projection on each transverse speckle plane (the 3D-SPFS) so-providing a 3D reconstruction of the object (Fig. 2d, e). Since the axial correlation length is shorter in the saturated regime as shown in Fig. 1j, k, super-resolution is also provided along the propagation axis of the beam. Line profiles of the beads along the axial coordinate (Supplementary Fig. 11) exhibit an average axial full width at half maximum $\delta z = 2.5 \pm 0.1\,\mu m$ in the nonsaturated regime and varies from $\delta z' = 1.7\,\mu m$ for the bottom bead closer to the coverslip, to $\simeq 2.0\,\mu m$ (two other beads) in the saturated regime. Axial resolution is here better for the bead closer to the coverslip for two main reasons: first, the index mismatch between the coverslip and the mounting medium (PVA) introduces aberrations to the speckle away from the coverslip. Second, the speckle spot was focused at the coverslip, thus exhibiting higher intensities, smaller speckle grains and steeper intensity gradients near the coverslip. For this reason, for 3D imaging of biological samples presented below, we carefully focused the median plane on the median plane of the 3D sample. For a bead lying at the surface of the coverslip, the axial resolution is measured by deconvolving a median-plane image by all the SPSF planes (Fig. 2m). In the saturated regime the correlation distance is reduced by a factor $\simeq 2$ as compared to the non-saturated case. We attribute the axial resolution improvement to the presence of nodal optical vortex lines dominating the contrast (and thus the cross-correlation product) in the saturated regime, while contrast is dominated by speckle grains in the linear excitation regime. The tilted trajectory of nodal vortex lines in 3D speckle patterns[39,40] improves axial resolution.

Next, a sample consisting in 100 nm fluorescent beads deposited on a coverslip was prepared to measure transverse resolution improvement under saturated excitation conditions. Speckle images are shown in Fig. 2f, i under nonsaturated and saturated excitation conditions, respectively. The experimental measurements of SPSFs in the nonsaturated and saturated cases (Fig. 2g, j, respectively) obtained using isolated 100 nm beads allow object reconstruction (Fig. 2h, k, respectively). The image retrieved from the saturated excitation condition clearly demonstrates a higher resolution power, resolving every individual bead. Some neighboring beads can only be resolved by saturating fluorescence excitation. Line profiles plotted in Fig. 2l show a resolution down to $\simeq 147$ nm in the saturated case. The width $w$ of the effective point spread function can be estimated by removing the contribution of the bead size $d$ from the width of the bead image $W$ according to $W = \sqrt{d^2 + w^2}$, so giving $w = 108$ nm: a factor 3.3 below the diffraction limit ($0.514\lambda/NA = 355$ nm for $NA = 0.77$). The size of the bead used for the SPSF characterization limits the utmost achievable resolution, but smaller fluorescent beads yielded too low signal to background ratios. For comparison, an image was also taken by scanning a diffraction-limited spot in the sample, after correcting aberrations of the system thanks to the SLM (Supplementary Fig. 12). As expected, even the resolution of the deconvolved point-scanning image is already outperformed by the image retrieved from the non-saturated speckle image by a factor $\simeq \sqrt{2}$. This improvement can be explained by the flatter optical transfer function of the speckle scanning microscope as compared to point-scanning one (see Supplementary Note 1 and ref. [41]).

**Imaging of vesicles in cultured cells**. To demonstrate the practical applicability of 3D super-resolution speckle microscopy to biological samples, we imaged lysosomes in fixed cultured HeLa cancer cells, immuno-labeled by targeting lysosomal associated membrane protein 1 (LAMP-1). Lysosomes are vesicles whose size vary in the range between 50 and 500 nm[42]. A region of

interest was first defined in a cell (Fig. 3e). Two 2D-speckle-images (nonsaturated and saturated) were recorded (as illustrated in Fig. 3a) and first Wiener deconvolved (Fig. 3b, left column). A point scanning image-stack (Fig. 3a) was also recorded and 3D-deconvolved with the point-scanning PSF (Fig. 3c). Regions of two 1 μm distant transverse planes are shown, where some nearby vesicles can only be resolved under saturated excitation conditions (see zoomed insets and white arrows in Fig. 3b). In speckle imaging, the depth-of-field is defined by the size of the SPSF. For Lysosomes imaging, a 5 μm SPSF was used providing a ~5 μm depth-of-field. In Supplementary Fig. 13, a more complex object, namely actin filaments, are imaged with a ~15 μm depth-of-field using a 10 μm speckle spot. The scaling of the depth-of-field with the speckle spot size and the numerical aperture is discussed in Supplementary Note 4.

**Reconstruction by compressed sensing**. Reconstruction by plane-by-plane Wiener deconvolution of the 2D speckle image exploits the statistical orthogonality of speckles lying in different axial planes. However, since speckles are only orthogonal in a statistical sense, out-of-focus point-sources contribute to a background noise whose amplitude is inversely proportional to the number of speckle grains (see Supplementary Note 4). In a worst-case scenario, a bright out-of-focus point-source may even theoretically blind a fainter one because of the out-of-focus reconstruction background. In Supplementary Note 4, we demonstrate that for a scanning speckle pattern containing $N$ speckle grains in each axial plane, the sample should not exhibit more than $\simeq \sqrt{N}$ point-sources to maintain a signal to noise ratio higher than 1 after cross-correlation projection. Wiener reconstruction is expected to require similar conditions. Moreover, even for an isolated point source, reconstruction by Wiener deconvolution yields a peak surrounded by noise. For these reasons, although vesicles can be distinguished, a significant background is observed in Wiener-reconstructed images in Fig. 3b.

Such reconstruction noise artifacts can be suppressed using compressed sensing algorithms. In this regard, our speckle scanning techniques probing the sample with random point spread functions, is ideally suited[21,24]. Here, a fast iterative shrinkage-thresholding algorithm (FISTA) was used[43,44]. This algorithm iteratively solves such linear inverse problem as ours, by additionally taking the sample sparsity into account. FISTA converges toward a Lagrangian minimization:

$$\min_x \{ F(\mathrm{x}) \equiv ||A\mathrm{x} - \mathrm{b}||^2 + \lambda ||\mathrm{x}||_1 \}, \qquad (3)$$

where $x$ designates the object's coefficients (here, in the voxel basis where lysosomes can be described with a minimum number of coefficients), $A$ is the random projection matrix (the 3D-SPSF), $b$ the experimental 2D speckle-image, and $\lambda$ a Tikhonov regularization parameter. According to compressed-sensing theory[21], one can reconstruct a K-sparse object (containing K nonzero coefficients) in a volume containing $N_x N_y N_z$ voxels if the number of independent random measurement points $M$ satisfies: $M \geq O[K \log(N_x N_y N_z)]$. The results obtained by FISTA are shown in Fig. 3b (right column) for comparison with Wiener deconvolution. As a result, reconstruction is achieved with a drastic reduction of noise. Removal of out-of-focus background is also evident on axial cross-sections shown in Fig. 3f where the two vesicles pointed out by white arrows in Fig. 3b, c are represented. As a results, axial resolution is better with FISTA reconstruction. However, transverse resolutions are similar in both reconstruction schemes, meaning that super-resolution can solely be attributed to optical saturation. A more complete description as well as a quantitative characterization of sample-sparsity requirements is given in Supplementary Note 5,

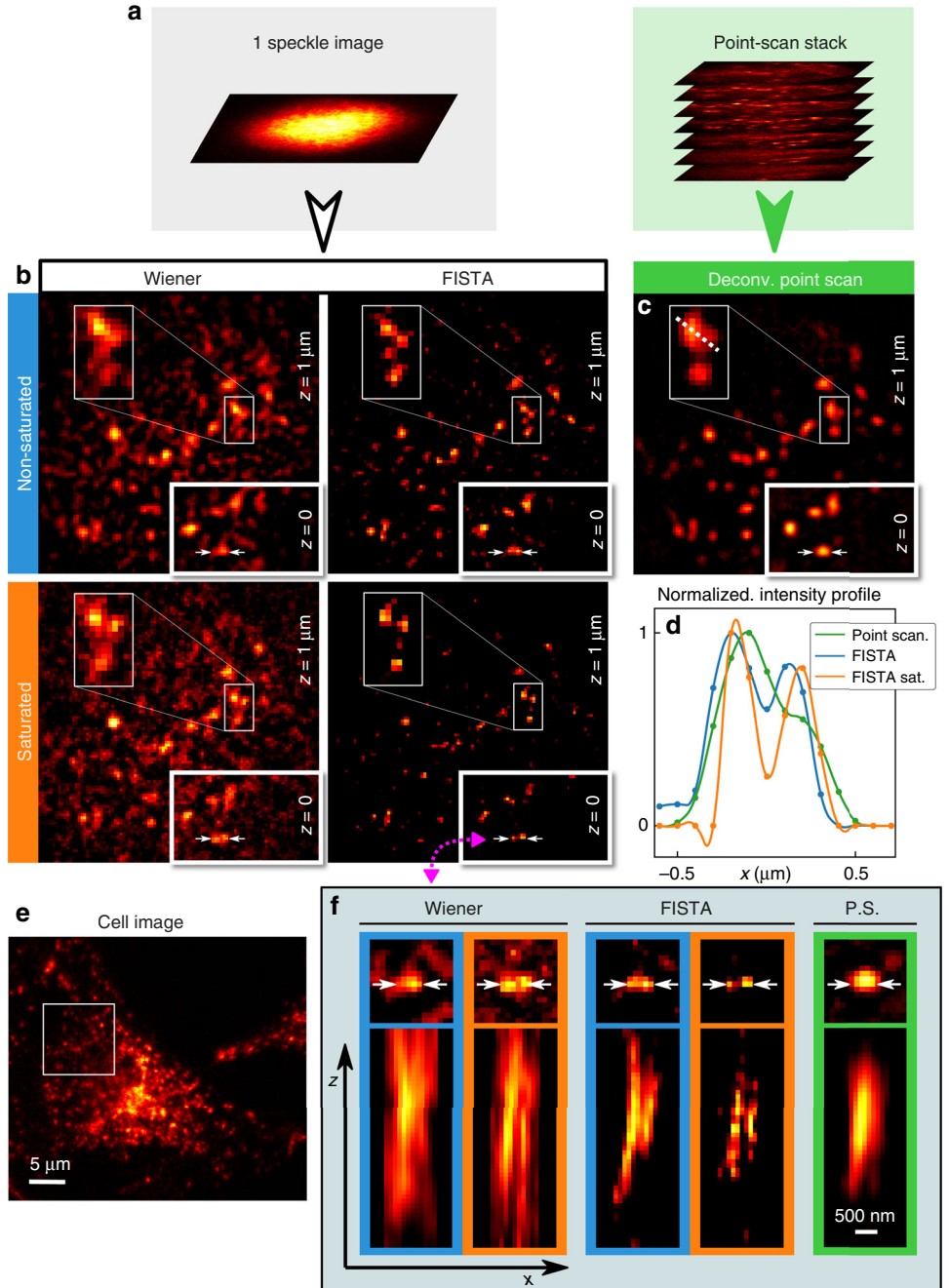

**Fig. 3** Images of lysosomes in fixed cultured cells. A region of interest materialized by a white square in the widefield view (**e**) was imaged both by 2D speckle scanning and 3D point scanning (**a**). Regions of two 1 μm distant transverse planes from the 3D reconstruction of fluorescence distribution are shown in (**b**, **c**), in three imaging modes: nonsaturated and saturated ($\langle s \rangle \simeq 1.4$) speckle imaging (**b**) as well as 3D deconvolved point scanning **c**. Reconstruction from speckle images is performed both by Wiener deconvolution and by the FISTA compressed sensing algorithm, for comparison. FISTA reconstruction suppresses noise arising from out-of-focus fluorescent objects. The line profile corresponding to the white dotted line in (**c**) is plotted in (**d**) (with quadratic interpolations) and compared to corresponding profiles obtained with FISTA reconstruction. White arrows at plane $z = 0$ pin-point two vesicles that can only be resolved under saturated conditions. Cross-sections of these two vesicles are shown in (**f**). All images were acquired using NA = 0.77

demonstrating high fidelity reconstruction by FISTA, even for much denser objects.

## Discussion

We demonstrated that speckle patterns are ideally suited to achieve compressed 3D microscopy beyond the diffraction limit under saturated fluorescence excitation conditions. Despite the vectorial nature of light waves which disallow perfect zeros of intensity in a random light-wave structure, 108 nm resolution could be obtained for a NA of 0.77, a factor 3.3 below the diffraction limit. Here, samples were imaged thanks to 2D raster scans but, in principle, for sparse-enough objects, reconstruction could be achieved with even fewer measurements. This technique, based on the sparsity of the sample, opens up new perspectives to perform super-resolution imaging with potentially drastic

acquisition-time reduction and photo-bleaching minimization. Contrary to regular SIM and confocal microscopy which wastes out-of-focus fluorescence signal, compressive 3D speckle imaging by 2D scanning makes use of all the detected fluorescence signal. Efficient compressive probing of 3D objects with a single 2D scanned image is ensured by the statistical orthogonality of random speckles. The cross-talk between different transverse planes can be suppressed by using a FISTA. Here, imaging of samples that are sparse in the voxel basis was achieved. Simple numerical deconvolution techniques could also be used for the sake of physical evidence both of 3D imaging and super-resolution capabilities. However, in practice, more complex objects could be imaged since all typical objects can be described with a sparse set of modes, provided a proper basis is used[45]. For objects not sparse in the voxel basis, the only limitation is the contrast of the 2D speckle image which must be larger than the photon shot-noise. Even for such objects, imaging with speckles allows efficient compressed sensing[25,46] since random structures are strongly incoherent (i.e., orthogonal) with all typical sparsifying bases[21].

The presented results were obtained with a very simple and inexpensive system, where the microscope objective could be easily changed for any high NA optics (like a condenser), whose optical properties are poor but which can efficiently collect the fluorescence signal. In our experiments, we observed that photo-bleaching (Supplementary Fig. 8) was less critical than background signal originating from the optics and the immersion oil (Supplementary Fig. 7). Here, background signal was important because fluorescence excitation was saturated, but 3D super-resolution speckle imaging could also be performed saturating other optical transitions such as stimulated emission, which makes use of red-shifted light as an intense laser beam and thus both reduces background signal and photo-bleaching. In this case, speckle patterns having inverted intensity contrast[47] could be used for the excitation and the de-excitation speckle patterns. Speckles with tailored statistical properties could also be used[48], such as non-Rayleigh speckles[49], potentially improving statistical orthogonality. Nondiffracting speckle beams[50] could also allow tuning the depth of field.

## Methods

**The speckle-scanning microscope.** A complete scheme of the experimental setup can be found in Supplementary Fig. 1. For saturated speckle imaging, the laser source is a 532 nm Q-switched laser diode delivering ~4 µJ, 500 ps pulses at 4 kHz (Teem Phononics, Meylan, France, STG-03E-120). A fully developed speckle pattern is generated by a SLM (Hamamatsu Photonics, LCOS, X10468-01) conjugated to the back focal plane of a 1.4 NA microscope objective (Olympus, Tokyo, Japan, 100×, NA 1.40, UPLanSApo, Oil) which focuses the beam into the sample. A quarter wave-plate is placed right before the objective lens to polarize the illuminating beam circularly. In addition, an iris placed before the SLM allows controlling the NA of the speckled beam. Finally, the speckle pattern is scanned transversely in the sample by a pair of galvanometric mirrors and fluorescence is then collected through the same objective lens and sent to a photo-multiplier tube (Hamamatsu Photonics, H10721-20). A pinhole limiting the field of view to ~10 µm was placed in an intermediate image plane between the objective and the photomultiplier tube to minimize background signal. The pixel dwell time corresponds to two laser pulses in 2D images and four pulses for the 3D images. In Fig. 1f, g, NA = 0.77, and for the sake of illustration clarity, in Fig. 1d, e, NA = 0.33. In Fig. 2f–k, 100 nm-beads and 50 nm pixel size were used for transverse resolution characterization. For axial resolution characterization, brighter 200 nm beads were used (in Fig. 2a, b) in order to help the experimentalist finding a region of interest on the camera. In Fig. 3, 2D speckle images were recorded using a 5 µm speckle spot with a 100 nm pixel size.

**Object reconstruction.** In Fig. 2d, e, plane-by-plane Wiener deconvolutions were performed. The speckled point spread functions were characterized by scanning isolated 100 nm fluorescent beads (200 nm for axial resolution characterization). For deconvolution of speckle images of Lysosomes in Fig. 3 (and actin filaments in Supplementary Fig. 13), both SPSFs were obtained by scanning distinct samples of fluorescent nano-beads. The saturated SPSF were estimated form these beads although the dye used was different. It appeared that the uncertainty about the exact saturation level is not critical for reconstruction. For Wiener deconvolution,

the spectral noise density was adjusted by visual inspection in order to optimize the compromise between resolution improvement and the signal to noise ratio. In Fig. 2, deconvolved images are shown without any threshold to show the noise level. FISTA algorithm has two tuning parameters: the sparsity degree $\lambda$ and the step size $t$. For optimal reconstruction, the step size $t$ is maintained as small as possible ($10^{-7}$) and $\lambda$ is varied between 0 and 1; 0 is optimal for least sparse objects and 1 for most sparse objects. The optimization is done by comparing the root mean squared error a posteriori on reconstructions using different parameters. Object reconstruction by FISTA typically took from roughly 4–6 min (371.58 s for $170 \times 170 \times 25$-voxels data and 226.54 s for $150 \times 150 \times 25$-voxels data), performing 2000 iterations (providing good resolution as shown in Fig. 3), on an Intel core i7-6700 processor, clocked at 3.4 GHz.

**Samples.** Fluorescent beads (FluoSpheres®, Molecular Probes, carboxylate-modified microspheres, 0.1 µm, orange fluorescent (540/560) 2% solid) were spin-coated in a PVA matrix on a coverslip and mounted in an anti-fade mounting medium (Fluoromount, Sigma-Aldrich). Cultured cancer HeLa cells grown on coverslips were washed in phosphate-buffered saline (PBS) and fixed with 4% paraformaldehyde. Then, they were incubated in blocking solution for 30 min at room temperature. Primary antibody targeting LAMP-1 were diluted in blocking buffer and incubate 1 h at room temperature. After several rinces, coverslip were incubated with fluorescent Alexa 555-conjugated secondary antibody for 45 min at room temperature, rinced extensively in PBS and mounted with Fluoromount antifading medium. The cell preparation and staining protocole followed the European Union and institutional guidelines and was validated by the Centre National de la Recherche Scientifique (CNRS). For actin samples, monomeric G-actin was polymerized by addition of 100 mM KCl and 2 mM MgCl$_2$[51]. Actin filaments were stabilized using Alexa Fluor 546 phalloidin (Thermo Fisher), and anchored on coverslips functionalized with inactive Myosin 1b[52,53]. Finally, the sample was mounted in Fluoromount after several washes. Samples of beads in 3D were prepared by simply drying a colloidal suspension of beads with PVA. The actin filaments in 3D were mounted in Fluoromount.

**Reporting summary.** Further information on experimental design is available in the Nature Research Reporting Summary linked to this article.

## Data availability
Data that support the findings of this study are available from http://www.biomedicale.parisdescartes.fr/doc/raw.zip.

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

## Acknowledgements

The authors thank Isabelle Fanget for preparing the HeLa cells, Julien Pernier for providing the actin filaments and the corresponding staining protocol, and Laura Caccianini, Madjouline Abou Ghali, and Dany Khamsing for helping with samples. The authors also acknowledge Grace Kuo and Laura Waller for their support with the FISTA algorithm, and Marcel Lauterbach for careful reading of the manuscript. This work was supported by grants from the Région Ile-de-France (PhD program) and the French-Israeli Laboratory ImagiNano. It has received funding from the European Research Council (ERC) under the European Union Horizon 2020 research and innovation program (Grants no. 677909) and from the French National Agency (ANR-10-INBS-04-01). This research was also supported by the ISRAEL SCIENCE FOUNDATION (Grant no. 1361/18), the Azrieli Foundation, the Centre National de la Recherche Scientifique and the France Bio-Imaging infrastructure.

## Author contributions

M.P. built up the experimental system, did experiments and numerical image processing. S.G. did experiments. A.T. and O.K. performed numerical image reconstruction by FISTA. O.K. and V.E. co-supervised the project. M.G. conceived and supervised the project. M.P., A.T., O.K. and M.G. wrote the paper.

## Additional information

**Competing interests:** The authors declare no competing interests.

**Journal Peer Review Information:** *Nature Communications* thanks the anonymous reviewers for their contribution to the peer review of this work. Peer reviewer reports are available.

