## [Peer Review File · Nature Communications]

Reviewers' comments:

Reviewer #1 (Remarks to the Author):

The authors present a way of using Nonlinear structured illumination microscopy to allow 3D super resolution. More specifically they project saturating fluorescence excitation with tightly focused speckle patterns to obtain the 3D improvement. They get the 3D improvement by doing a single two-dimensional scan of the sample which exploits both the natural axial orthogonality of speckle illumination and the sparsity of the sample.

In general, the idea of obtaining 3D super resolution is applicable and important. The concept the authors apply is also valid and should work. The manuscript is well written, being very clear and didactic in its structure. I have two main problems with this work:

- It is true that moving towards 3D is novel but the approach in general has already been done before and thus the novelty is mainly involved in the improved usage of speckles, sparsity and the scanning procedure.
- The results presented don't show very significant improvement. The authors claim for improvement factor of 2.3 beyond the diffraction barrier which is relatively small improvement and in the presented images the improvement seem to be even less evident.

I therefore cannot recommend accepting the paper for publication in its present form before the authors will do a major revision. The revision should in my opinion related to the following points which were not clear or disturbed me in their work:

- Getting more impressive results especially using real biological samples, e.g. cells etc.
- Commenting on the problem the proposed concept may have when being applied for biological applications particularly in respect to the time it will take to do the axial scan and the might intensity the concept may require while both can be problematic for living and moving entities.
- When applying the concept on biological samples there is a scattering process which will change the random speckle patterns. How the authors aim to overcome this in their decoding procedure? How will they separate between primary and secondary speckle patterns generated in the biological medium when applying their approach?

Reviewer #2 (Remarks to the Author):

Review on '3D super-resolution microscopy ...' by Pascucci et al

There are two main ideas in this manuscript which are, in my opinion, independent.

In a first part, the authors describe a super-resolution fluorescence microscope using saturated-speckle scanning illumination. It shows that a resolution about 150 nm can be obtained with a subnanosecond pulsed laser with wavelength 532nm and a numerical aperture about 0.77. Unfortunately, increasing the NA does not permit to improve the resolution as the non-zero axial component of the speckle field prevents an efficient saturation of the fluorophore optical transition.

In a second part, they show that speckle illumination can provide a 3D image from a simple 2D scan, using the orthogonality of the illumination on transverse planes that are sufficiently far apart. Note that this technique works only on sparse samples for avoiding being blinded by the

out-of-focus fluorescence. The 3D imaging from 2D scanning is demonstrated experimentally on actin filament using non saturated speckles. A 3D imaging using saturated speckles is performed on a simpler sample made of 3 beads (which is not as convincing).

I found both ideas interesting (despite their limitations) and worth being published. However, I think that in its present state, the manuscript is unclear because it refers too many times to the supplementary material. The latter is absolutely necessary to the understanding of the main text and is quite long !!.

In my opinion, this manuscript could be split in two independent papers, one on the saturated speckle and the other one on the 2D to 3D imaging using speckle illumination (possibly non saturated). The supplementary material could be re-introduced in these two papers for improving their readability.

Apart from this important suggestion, I have a few comments concerning the first idea.

1) The speckle scanning technique requires deconvolution. Two techniques are presented, one which requires a prior recording of the speckle pattern (in 2D or 3D), one which does not require to know the speckle pattern (provided that the latter is translated without modification in the sample). The blind deconvolution has been applied only to the 2D configuration with $NA=0.33$ (from what I have understood). Is there a reason for this ? Is this technique, based on the autocorrelation of the image, less efficient than a direct deconvolution using the known pattern ? In addition, how is this direct deconvolution sensitive to any error on the pattern ? Is this sensitivity increased when dealing with saturated speckle ?

2) Is the super-resolved image obtained with two different speckle patterns show some discrepancy after deconvolution ? Would it be interesting to record several images of the same sample with different speckles in order to lessen the sensitivity to a specific speckle pattern ?

3) How is chosen the translation step of the pattern (for the saturated and non saturated configurations)?

4) It is not clear to me in which cases the saturated speckle scanning technique can be more interesting than the non saturated speckle scheme (using higher NA) or the classical saturated periodic SIM (as presented by Gufstasson) or the focused beam approach (STED) (apart from the simplification of the experimental set-up). Maybe the authors could comment this point.

Concerning the second idea, I think that the authors could refer to the work on the diffuserCAM recently presented by Laura Waller's team where 3D images are obtained from 2D data using a diffuser in front of a camera.

Reviewer #3 (Remarks to the Author):

Authors of "Three-dimensional super-resolution microscopy with speckle-saturated fluorescence excitation" describe a way to achieve 3D super-resolution imaging using saturated speckle patterns as the structured illumination. While this technique has been already shown to increase transversal (XY) resolution, to date no demonstration was made of three-dimensional super-

resolution. Here, authors perform 3D imaging of sparse samples with axial super-resolution by only scanning the illumination pattern in 2D transversely, using independence of speckle patterns at different axial planes (separated enough to decorrelate the speckle patterns).

I have found this paper methodology to be very explanatory and thoroughly written, including the supplementary material. However, some major points about the results need to be addressed before one could decide on its publication:

— Authors state that they pre-record 3D SPSFs (Fig. 4c), which are then used to deconvolve the images. How stable such SPSFs are and could one "re-use" them for different samples? Or is it then a requirement to embed sparse beads into every sample to measure the 3D SPSF? To put it in another way, how sensitive is the PSF to the sample preparation? How sensitive is it to slight changes in aberrations introduced by the glass slides or mounting medium?

— Could the authors provide numbers regarding the largest volume that could be imaged with their technique? From figure 4d (top), it looks like they can image a volume which is only 2 to 3 times thicker than the depth of field of the objective.

— In Fig. 3, authors show actin filament which is extending only few nanometers in the axial direction; when demonstrating 3D imaging of actin filaments in Fig. 4d, they demonstrate it only in case of non-saturated illumination, thus with worse resolution. The explanation the authors provide is that the fluorescence background given by the mounting medium prevent them to use saturated speckles in this specific case.

This is quite a downside, and I think the authors should at least discuss what are the requirements in terms of absorption/emission of the object compared to the background.

It would help the reader to evaluate how useful the proposed technique could be for less controlled samples, as it adds a second constraint on top of sparsity.

— Why did authors use 200 nm beads in Fig. 4 as compared to the previous measurements (100 nm)? Was it only necessary to provide a brighter fluorescent emission, or were there other reasons?

— The authors claim that super resolution is due to optical vortices in the speckle pattern. In this regard, they take special care to circularly polarize the impinging random speckle field to ensure an isotropic enhanced resolution.

I would be curious to see how much the resolution is improved by only saturating the gaussian psf used for point scanning (with identical saturation conditions), as it also introduces higher spatial frequencies. The experiment can easily be realized with the existing setup, and would help to assess what is the exact role of the optical vortices in the speckle pattern, and quantify its importance for the resolution improvement.

— In Figure 2i the authors show line profiles of imaged 100nm beads to assess the resolution of their imaging system — with FWHM of 147 nm for the saturated illumination.

In the main text associated with this figure the "effective PSF" is claimed to be 100 nm and 3.4 times below the diffraction limit. What is being used as the metric for the "effective PSF" in this case?

In the introduction, the increase in resolution is estimated to be 2.3 fold. I could obtain this value by comparing the diffraction limit of their objective to the 147 nm bead line profile FWHM.

However, this metric is flawed, since they compare FWHM of the line profile in case of the saturated illumination to the diffraction limit as $\lambda/2NA$. For fair comparison, it seems that FWHM of the Airy disk should be used ($0.4 \lambda/NA$) which changes the resolution increase factor to ~ 1.88 .

It would be beneficial if the authors could in general state more clearly how they obtain all these numbers, i.e. how they actually define their resolution.

— Could the authors provide axial cross-sections of the beads in figure 4d (top)? It would be

easier to visualize the resolution increase.

– Please be careful about typos, in the main text (orthogonal, arising), as well as in the supplementary material (fluorescence).

– Some references are listed but not mentioned in the paper itself, mostly from reference 33.

Responses to reviewers' comments:

We thank the reviewers for their insightful reading of our manuscript and for their constructive criticisms. We addressed all the points they raised and think that the manuscript has now significantly improved. Point by point responses are appended below in black ink.

Reviewer #1 (Remarks to the Author):

The authors present a way of using Nonlinear structured illumination microscopy to allow 3D super resolution. More specifically they project saturating fluorescence excitation with tightly focused speckle patterns to obtain the 3D improvement. They get the 3D improvement by doing a single two-dimensional scan of the sample which exploits both the natural axial orthogonality of speckle illumination and the sparsity of the sample.

In general, the idea of obtaining 3D super resolution is applicable and important. The concept the authors apply is also valid and should work. The manuscript is well written, being very clear and didactic in its structure.

We thank the reviewer for pointing out the importance of this work and for his positive criticisms about the clarity of the manuscript. Responses to the reviewer's issues are appended below.

I have two main problems with this work:

- It is true that moving towards 3D is novel but the approach in general has already been done before and thus the novelty is mainly involved in the improved usage of speckles, sparsity and the scanning procedure.

We fully acknowledge that the principle of the orthogonality of random or out-of-focus speckles has been known for a long time. However, it has never been applied for achieving three-dimensional microscopy and we think that the novel suggested scanning and reconstruction approaches are of considerable interest for the microscopy community. In our new manuscript, we further stressed on the synergetic combination with optical saturation for super-resolution imaging, making use of all detected fluorescent photons, in contrast with regular SIM approaches.

- The results presented don't show very significant improvement. The authors claim for improvement factor of 2.3 beyond the diffraction barrier which is relatively small improvement and in the presented images the improvement seem to be even less evident.

Based on the response to reviewer 3, we actually obtain a 3.3 point spread function reduction. The routinely used linear structured illumination microscopy only allows a 2-fold improvement in resolution also without offering the advantages of rapid 3D scanning. We understand that the reviewer may expect better resolutions because saturated microscopy techniques in principle allow breaking the diffraction barrier up to theoretically unlimited resolution. Here, our resolution improvement is solely limited by the signal to noise because of background signal under saturated excitation conditions by intense laser pulses. However, the same principle could be applied by generating less background signal: namely saturating other photo-physical mechanisms such as optical photo-switching or stimulated emission

depletion would induce much less fluorescence in the spectral-range of the detection-window. A sentence was added in the discussion to open on this perspective:

“ In our case, background signal was important because fluorescence excitation was saturated but 3D super-resolution speckle imaging could also be performed saturating other optical transitions such as stimulated emission, which makes use of red-shifted light as an intense laser beam and thus both reduce background and photo-bleaching.”

I therefore cannot recommend accepting the paper for publication in its present form before the authors will do a major revision. The revision should in my opinion related to the following points which were not clear or disturbed me in their work:

- Getting more impressive results especially using real biological samples, e.g. cells etc.

We thank the reviewer for his suggestion. We now better discuss the resolution improvement and imaged biological samples consisting in lysosomes in cultured cells which, we think, will definitely strengthen the impact our manuscript.

- Commenting on the problem the proposed concept may have when being applied for biological applications particularly in respect to the time it will take to do the axial scan and the might intensity the concept may require while both can be problematic for living and moving entities.

We think that the experimental evidence obtained by imaging cultured cells satisfactorily responds to the reviewer concerns about the required signal's intensity. We are not sure to understand what the reviewer means about the axial scanning of the samples since the strength of our technique is precisely to only require a single 2D scan. Only the point spread function must be characterized in 3D.

- When applying the concept on biological samples there is a scattering process which will change the random speckle patterns. How the authors aim to overcome this in their decoding procedure? How will they separate between primary and secondary speckle patterns generated in the biological medium when applying their approach?

We understand the reviewer expectation about efficient 3D super-resolution imaging in scattering samples. Imaging deep in scattering tissues will definitely change the speckle. Our work does not claim tackling this topic which falls outside the scope of this manuscript. Therefore, we payed attention to remove from the manuscript any item that could confuse the reader.

Reviewer #2 (Remarks to the Author):

There are two main ideas in this manuscript which are, in my opinion, independent.

We thank the reviewer for this criticism and understand the reviewers point. We agree that in the former version, the synergetic link between 3D imaging and super-resolution was not clear enough. Therefore, we significantly changed the structure of the manuscript to better highlight the importance of

compressive 3D speckle imaging which not only concern acquisition time but also just allow 3D super-resolution by saturated fluorescence excitation, which has never been achieved before. The reason is that regular structured illumination microscopy discard out-of-focus fluorescence signal by numerical sectioning and requires so many acquisitions as the number of transverse planes. In contrast, our system makes use of all collected photons and allows 3D imaging in a single 2D-image. We now think that these changes significantly improve the impact of our manuscript.

In a first part, the authors describe a super-resolution fluorescence microscope using saturated-speckle scanning illumination. It shows that a resolution about 150 nm can be obtained with a subnanosecond pulsed laser with wavelength 532nm and a numerical aperture about 0.77. Unfortunately, increasing the NA does not permit to improve the resolution as the non-zero axial component of the speckle field prevents an efficient saturation of the fluorophore optical transition.

The reviewer is wright stressing on the importance of controlling the axial component of the field. However, we would like to first stress on the fact that the 0.77NA is not so weak since x40 or x60 objectives with 0.8 or 0.9 NA are typically used by biologists. Second, the axial component is only non-zero when averaging over vortices. Therefore, increasing the NA will only reduce the fraction of “dark-enough” vortices. The consequence is a reduction of the signal-to-noise ratio but not a suppression of super-resolved information. In this regard, saturating fluorescence excitation is particularly critical since inducing a significant amount of background. In the future, saturating stimulated emission rather than fluorescence excitation would improve the signal to noise ratio and should thus improve the usable NAs. We thus do not share the pessimistic *a priori* opinion of the reviewer about the impossibility to increase the NA in the future.

Following these considerations, we modified the introduction to bring this precision:

“The vectorial nature of light cannot be neglected in tightly focused beams, especially under saturated conditions, in which case the axial field may destroy **most of** intensity zeros.”

In a second part, they show that speckle illumination can provide a 3D image from a simple 2D scan, using the orthogonality of the illumination on transverse planes that are sufficiently far apart. Note that this technique works only on sparse samples for avoiding being blinded by the out-of-focus fluorescence. The 3D imaging from 2D scanning is demonstrated experimentally on actin filament using non saturated speckles. A 3D imaging using saturated speckles is performed on a simpler sample made of 3 beads (which is not as convincing).

As explicitly suggested by Reviewer 1 and implicitly by Reviewer 2, we now imaged Lysosomes in cultured cells to demonstrate the possibility to achieve 3D super-resolution imaging of a typical biological sample. We feel that this experimental demonstration now makes our technique perfectly suitable for biology. To answer the reviewer’s concern, we modified the manuscript the following way:

- First, while conventional structured illumination microscopy wastes out-of-focus fluorescence by numerical sectioning, our technique makes use of this signal. We now make this point clear in the introduction:

“Imaging in 3D is then obtained by acquiring every single transverse plane. In this configuration, fluorescence arising from out-of-focus planes is thus induced to be wasted since suppressed by numerical sectioning [19]”

And in the conclusion:

“Contrary to regular structured illumination microscopy and confocal microscopy which wastes out-of-focus fluorescence signal, compressive 3D speckle imaging by 2D scanning makes use of all the detected fluorescence signal.”

- Second, we now introduced a new algorithm allowing suppressing out-of-focus fluorescence by using FISTA, a compressive sensing algorithm. The third part of the Results section (p.8) now extensively discusses the problem of out-of-focus fluorescence as well as this new algorithm with experimental evidence of reconstruction improvement in Fig. 3.

- Third, as suggested by the reviewer, we now cite the sparsity based algorithm by Laura Waller as a solution in several occurrences in the manuscript. The reviewers' suggestion was all the more relevant that the proposed algorithm in ref. 26 would indeed efficiently remove this out-of-focus contribution. Ref. 26 is now cited in three places.

In the introduction:

“This property exactly provides the random-projection-measurement configuration ideally suited for compressed imaging reconstruction [24,25], in particular for 3D imaging [26]” (p. 2)

In the “Results” section:

“The resulting image is thus the linear combination of the contribution of all the planes. Since speckles are “quasi-orthogonal”, each planes can then be retrieved either by projection [23] or compressive sensing schemes [26]”

“Adding sparsity constraints to the rebuilt object is particularly helpful, especially to retrieve 3D information from 2D images [26]”

I found both ideas interesting (despite their limitations) and worth being published.

We thank the reviewer for his positive appreciation of our work. We definitely feel that the manuscript has now been further improved thanks to the reviewers' comments.

However, I think that in its present state, the manuscript is unclear because it refers too many times to the supplementary material. The latter is absolutely necessary to the understanding of the main text and is quite long !!. In my opinion, this manuscript could be split in two independent papers, one on the saturated speckle and the other one on the 2D to 3D imaging using speckle illumination (possibly non saturated). The supplementary material could be re-introduced in these two papers for improving their readability.

We thank the reviewer for stressing on the scientific interest of the supplementary materials and for his

suggestion to split our work into two different papers. However, as argued above, we hope it is now clearer that there is a real synergetic link between 3D speckle imaging and 3D super-resolution. We feel this imaging scheme is of considerable interest for the community. Although the reviewer's interest apparently urged them to spend time to understand our work in deep details (and we are very grateful for their interest), we paid attention to only place in supplementary materials theoretical demonstrations as well as supplementary experimental data, keeping the main results in the manuscript for the sake of readability. We thank again the reviewer for his insightful reading and hope that they will appreciate the current version of the manuscript.

Apart from this important suggestion, I have a few comments concerning the first idea.

1) The speckle scanning technique requires deconvolution. Two techniques are presented, one which requires a prior recording of the speckle pattern (in 2D or 3D), one which does not require to know the speckle pattern (provided that the latter is translated without modification in the sample). The blind deconvolution has been applied only to the 2D configuration with $NA=0.33$ (from what I have understood). Is there a reason for this ?

All the results presented in the article require characterizing the 3D speckle point spread function. Blind deconvolution, such as shown in supplementaries (Fig. S15), has only been applied to 2D configuration because the current state of the art of phase-retrieval algorithms does not work in the novel 3D configuration we suggest. Unless adding assumptions about the 3D structure of the scanning speckle, the algorithm cannot discriminate between the different transverse planes. We agree that developing algorithms working for our 3D configuration would be of very high interest in the future. Using $NA=0.33$, allowing larger resolution improvement factor than $NA=0.77$, it demonstrates that super-resolved information can definitely be retrieved by the blind reconstruction algorithm. We kept the blindly reconstructed image in supplementary data but we now removed any reference to this experimental result in the main text to avoid confusion about the main topic. We feel that discussing the topic of blind reconstruction algorithms would make the topic too vast. We rephrased the sentence mentioning the possibility to use blind reconstruction algorithms:

“ Techniques have been developed to reconstruct images even when the speckles are unknown [15, 17] especially through scattering samples [33-36]”

Is this technique, based on the autocorrelation of the image, less efficient than a direct deconvolution using the known pattern ?

Blind deconvolution algorithms are indeed less robust to noise. Moreover, they involve several *ad-hoc* parameters. As written in the manuscript, our goal is here to demonstrate first that super-resolution is obtained by the power-spectrum enlargement under optical saturation by reconstruction with simple Wiener deconvolution and second, to demonstrate that a more elaborated compressed sensing algorithm (FISTA) can get more reliable results. As a supplementary result, we found it was interesting to demonstrate that similar resolution improvement could be obtained with a blind-reconstruction algorithm in a simple case. As answered to the former question, references to blind reconstruction as almost removed from the manuscript to avoid confusion.

In addition, how is this direct deconvolution sensitive to any error on the pattern ? Is this sensitivity increased when dealing with saturated speckle ?

The reviewer raises here an interesting but difficult question that should probably deserve a specific quantitative characterization. Our observation is that a perturbation in the speckle pattern degrades the signal to noise at reconstruction as well as resolution, especially under saturated illumination conditions. We noticed this effect when imaging the actin filament: if the speckle point spread function is not characterized in the very same plane as the speckle image, reconstruction fails. The problem of the degradation of the speckle pattern because of the index mismatch is discussed about the three beads. As suggested by reviewer 3, axial line profiles of the three beads have now been plotted in Fig. S10 and exhibit a better resolution improvement for the bead closer to the coverslip. However, reconstruction of the Lysosomes over a depth of field of $\sim 5\mu\text{m}$ illustrate that the index mismatch does not critically prevent either reconstruction or super-resolution capabilities.

A sentence about new Fig. 2e states this effect:

“Axial resolution is here better for the bead closer to the coverslip for two main reasons: - first, the index mismatch between the coverslip and the mounting medium (PVA) introduces aberrations to the speckle away from the coverslip - second, the speckle spot was focused at the coverslip, thus exhibiting higher intensities, smaller speckle grains and steeper intensity gradients near the coverslip.”

We agree that a quantitative discussion about optical aberrations could be interesting, relying for instance on numerical simulations of wavefront distortions by Zernike polynomials, but we feel that such an extensive study of the effect of optical aberrations would not fit in the scope of the manuscript.

2) Is the super-resolved image obtained with two different speckle patterns show some discrepancy after deconvolution? Would it be interesting to record several images of the same sample with different speckles in order to lessen the sensitivity to a specific speckle pattern ?

Changing the speckle pattern only changes the noise after reconstruction. For a speckle point spread function (SPSF) containing N speckle grains, the signal to noise ratio associated with Wiener deconvolution equals \sqrt{N} (Section S.3.3). In the case of a reconstruction by a compressed sensing algorithm, we think it would not change the end result. Changing the SPSF only changes the background signal and the depth of field. Recording several images of the same sample with different speckles would improve the signal to noise ratio at reconstruction, in a similar way to increasing the number of speckle grains in the speckle psf.

3) How is chosen the translation step of the pattern (for the saturated and non saturated configurations)?

In the non-saturated regime, the translation step is chosen in agreement with Shannon requirements, a fraction of the speckle grain size: at least a factor two below the Rayleigh criterion. For $\text{NA}=0.77$, $\lambda/(2\text{NA})=345\text{nm}$. For Fig. 2 f-k, the pixel size was chosen to be 50nm for resolution improvement characterization (a factor ~ 3 below the bead size image). For Lysosomes imaging, since resolution is slightly degraded by optical aberrations due to the index mismatch, we observed experimentally that an

optimal pixel size was 100nm. This information was added in the Methods section.

4) It is not clear to me in which cases the saturated speckle scanning technique can be more interesting than the non saturated speckle scheme (using higher NA) or the classical saturated periodic SIM (as presented by Gufstasson) or the focused beam approach (STED) (apart from the simplification of the experimental set-up). Maybe the authors could comment this point.

Performing 3D imaging with a 2D scan is interesting for several reasons: for acquisition speed as well as for photo-toxicity and photobleaching. It is then more interesting than techniques requiring 3D scanning (STED) or widefield techniques requiring so many recordings as the number of imaged planes (SIM). The proposed approach requires using speckles to ensure the orthogonality of defocused planes. Grids do not satisfy this requirement and do not allow compressive sensing. All-the-more, traditional SIM techniques and confocal microscopy (and STED under single-photon excitation) waste the out-of-focus fluorescence photons by numerical and physical optical sectioning, respectively. The limited photon budget available from fluorescent probes is thus spoiled.

We agree that resolutions obtained are not so good as those obtained by SIM and STED. However, the reason for this limit is due to limited signal to noise ratio under saturated excitation. The latter is further degraded by the axial component of the field at high NA and high saturation levels since it reduces the number of “useful” vortices. In the future, saturating other optical transition than fluorescence excitation such as stimulated emission (as suggested in the conclusion) is expected to improve the signal to noise ratio and thus change the response to the reviewer’s questions.

Concerning the second idea, I think that the authors could refer to the work on the diffuserCAM recently presented by Laura Waller’s team where 3D images are obtained from 2D data using a diffuser in front of a camera.

We thank the reviewer for this judicious suggestion. It is all the more relevant that using a compressed sensing algorithm now demonstrate significantly better results with reduced noise related to out-of-focus point-sources contribution. This work is now cited as ref. 26 in three different places in the manuscript as detailed above.

Reviewer #3 (Remarks to the Author):

Authors of “Three-dimensional super-resolution microscopy with speckle-saturated fluorescence excitation 1D” describe a way to achieve 3D super-resolution imaging using saturated speckle patterns as the structured illumination. While this technique has been already shown to increase transversal (XY) resolution, to date no demonstration was made of three-dimensional super-resolution. Here, authors perform 3D imaging of sparse samples with axial super-resolution by only scanning the illumination pattern in 2D transversely, using independence of speckle patterns at different axial planes (separated enough to decorrelate the speckle patterns).

I have found this paper methodology to be very explanatory and thoroughly written, including the supplementary material.

We thank the reviewer for his positive comment about our work.

However, some major points about the results need to be addressed before one could decide on its publication:

- Authors state that they pre-record 3D SPSFs (Fig. 4c), which are then used to deconvolve the images. How stable such SPSFs are and could one "re-use" them for different samples? Or is it then a requirement to embed sparse beads into every sample to measure the 3D SPSF? To put it in another way, how sensitive is the PSF to the sample preparation? How sensitive is it to slight changes in aberrations introduced by the glass slides or mounting medium?

Experimentally, the speckle point spread functions (SPSFs) were characterized with fluorescent beads and then allowed imaging actin filaments, other sample of beads or the cultured cells sample. Depending on the stability of the setup, the SPSF can be re-used for several samples. We now made it clear that there is no need to have a bead (as a guide star) in the imaged sample in the Methods:

"For deconvolution of speckle images of Lysosomes in Fig.3 (and actin filaments in Fig. S13), both SPSFs were obtained by scanning distinct samples of fluorescent nano-beads. The saturated SPSF were estimated from these beads although the dye used was different."

However, as answered to reviewer 2, we noticed that the index mismatch at the coverslip indeed alters the reconstruction quality, especially far away from the coverslip. The sensitivity of reconstruction under speckle point spread function (SPSF) perturbation would probably deserve a quantitative discussion based on numerical simulations. However, we do feel that such an extensive study would not fit in our already rich manuscript. As suggested by the reviewer, axial line profiles of the three beads have now been plotted in Fig. S11 and exhibit a better resolution improvement for the bead closer to the coverslip. A sentence was also added in the main text to summarize the result of Fig. S11:

"Line profiles of the beads along the axial coordinate (Fig. S.11) exhibit an average axial full width at half maximum $\delta z = 2.5 \pm 0.1 \mu\text{m}$ in the non-saturated regime and varies from $\delta z' = 1.7 \mu\text{m}$ for the bottom bead closer to the coverslip, to $\sim 2.0 \mu\text{m}$ (two other beads) in the saturated regime."

A sentence about new Fig. 2e also states resolution degradation with the distance from the coverslip:

"Axial resolution is here better for the bead closer to the coverslip for two main reasons: - first, the index mismatch between the coverslip and the mounting medium (PVA) introduces aberrations to the speckle away from the coverslip - second, the speckle spot was focused at the coverslip, thus exhibiting higher intensities, smaller speckle grains and steeper intensity gradients near the coverslip"

- Could the authors provide numbers regarding the largest volume that could be imaged with their technique? From figure 4d (top), it looks like they can image a volume which is only 2 to 3 times thicker than the depth of field of the objective.

We understand that the reviewers question about the "volume" is probably actually related to the depth

of field, since limitations in terms of transverse scanning are only just related to the optics and the scanning device.

- First the depth of field scales as the transverse size of the speckle point spread function (SPSF). Typically, the number of different transverse speckles in the SPSF is equal to \sqrt{N} , with N the number of speckle grains in the SPSF. This discussion is in the supplementary materials at paragraph S.3.3. Extended depth of field can thus be obtained either by reducing the numerical aperture, or, more interestingly, by increasing the size of the SPSF which also increases the number of transverse speckle planes and the number of accessible planes. In our experiments, we used SPSF of $5\mu\text{m}$ (Fig 3) and $10\mu\text{m}$ (Fig. 2) in diameter, giving access to depth of field of $\sim 5\mu\text{m}$ and $\sim 10\mu\text{m}$, respectively. In former fig 4d top, the 3 beads are not spanning over the full depth of field. In former Fig 4d bottom, a $15\mu\text{m}$ axial range can clearly be imaged with a $10\mu\text{m}$ speckle spot. A sentence was added to the manuscript to explain this point:

“For Lysosomes imaging, a $5\mu\text{m}$ SPSF was used providing a $\sim 5\mu\text{m}$ depth-of-field. In Fig. S.13, an actin filament image with a $\sim 15\mu\text{m}$ depth-of-field is obtained using a $10\mu\text{m}$ speckle spot. The scaling of the depth-of-field with the speckle spot size and the numerical aperture is discussed in Sec. S. S.3.3.” (p.8)

- Second, the number of transverse planes also depends on the degree of sparsity of the imaged object. A full discussion was added when using a compressed sensing algorithm for Fig. 3 and an extensive study of the reconstruction ability by FISTA algorithm was added in the supplementary materials S.3.4 depending on the sparsity degree of the sample and the signal-to-noise ratio of the data.

- In Fig. 3, authors show actin filament which is extending only few nanometers in the axial direction; when demonstrating 3D imaging of actin filaments in Fig. 4d, they demonstrate it only in case of non-saturated illumination, thus with worse resolution. The explanation the authors provide is that the fluorescence background given by the mounting medium prevent them to use saturated speckles in this specific case.

This is quite a downside, and I think the authors should at least discuss what are the requirements in terms of absorption/emission of the object compared to the background.

It would help the reader to evaluate how useful the proposed technique could be for less controlled samples, as it adds a second constraint on top of sparsity.

We probably misexpressed about the origin of the background signal. Here, for the 3D sample of actin filament, the preparation protocol consisted in staining the actin monomer first and polymerize it, next. Therefore, this protocol did not allow us to rinse the extra-dye from the mounting medium of the stained actin filaments. The mounting medium then contained some dyes. The mounting medium itself is not incriminated here, only the specific preparation protocol of the actin filaments. The mentioned figure has now been moved as supplementary data, and has advantageously been replaced by 3D super-resolved images of real biological samples. As demonstrated with imaging of cultured cells, samples prepared with standard staining protocols allowing rinsing after labeling works perfectly fine.

Furthermore, background signal originating from out-of-focus point source is now extensively discussed in the third part of the results (p.8) as well as in the supplementaries (S.3). We now use a compressed

sensing algorithm that suppresses out-of-focus fluorescence by taking the sample sparsity into account. Evidence of reconstruction improvement is shown in Fig. 3d.

- Why did authors use 200 nm beads in Fig. 4 as compared to the previous measurements (100 nm)? Was it only necessary to provide a brighter fluorescent emission, or were there other reasons?

For this figure, we needed to find specifically a structure consisting in several beads in different axial planes. We took 200nm beads for a trivial experimental reason. Just for selecting a proper region of interest using bright 200nm beads make the task much easier. Tracking out-of-focus 100nm beads in 3D on an 8-bit camera was almost impossible because of too dim signal. Since the three beads now serves as a pedagogical illustration for 3D imaging capabilities by 2D scanning as well as for axial resolution improvement under saturated excitation, using 200nm beads – much smaller than the axial resolution – is perfectly fine. However, we agree that this detail may be confusing for the reader and we moved this information to the methods with the corresponding explanation:

“ For axial resolution characterization, brighter 200 nm beads were used (in Fig. 2a and b) in order to help the experimentalist finding a region of interest on the camera.”

- The authors claim that super resolution is due to optical vortices in the speckle pattern. In this regard, they take special care to circularly polarize the impinging random speckle field to ensure an isotropic enhanced resolution.

I would be curious to see how much the resolution is improved by only saturating the Gaussian psf used for point scanning (with identical saturation conditions), as it also introduces higher spatial frequencies. The experiment can easily be realized with the existing setup, and would help to assess what is the exact role of the optical vortices in the speckle pattern, and quantify its importance for the resolution improvement.

The reviewer asks here an interesting and perfectly legitimate question that we also considered. However, we think that comparing super-resolution obtained by speckle or by a diffraction limited focused spot would not make sense anymore in the new version of the manuscript since we now better stress on the specific interest of performing compressive 3D imaging by 2D scanning in the specific context of saturated excitation. Our compressive sensing approach indeed allows making an efficient use of fluorescent photons, unlike regular confocal or structured illumination microscopy. The proposed approach requires using speckles to ensure the orthogonality of defocused planes. Focused points and grids do not satisfy this requirement and do not allow compressive sensing. Traditional SIM techniques and confocal microscopy (and STED under single-photon excitation) waste the out-of-focus fluorescence photons by numerical and physical optical sectioning, respectively. The limited photon budget available from fluorescent probes is thus spoiled. We understand from the reviewer's question that this synergetic combination was not clear and we hope that this new structure answers the reviewer's question.

Now, to specifically answer the reviewer's question, from an experimental point of view, this question is not so trivial as it first appears since point scanning is not really achieved using a Gaussian psf but an Airy disk. A Gaussian beam has indeed the specific property to not contain any zero intensity point but this property disappears immediately in the presence of an iris (the pupil of the microscope objective) or optical aberrations. In our system, an iris is used to precisely control the illumination NA. Consequently, unlike a Gaussian spot which does not contain any zero, the Airy spot contains many concentric zero circles associated with edge phase dislocations (vortices are spiral phase dislocations). Moreover, edge phase dislocations are unstable structures, meaning that under infinitely small perturbation (like aberrations) they break into rosaries of vortices. An illustration of an experimental saturated Airy spot psf, asked by the reviewer, is shown in the figure below (scale bar=1 μ m). Therefore, it is almost impossible experimentally to discriminate this way between the role of zeros and the role other bright structures in resolution improvement.

Moreover, both in regular saturated structured illumination microscopy and in STED microscopy, it is clear that super-resolution is achieved thanks to the presence of dark structures (otherwise, the contrast would vanish). In our case, it is also clear that using a low-contrast speckle would extinguish contrast under saturation conditions. In Fig 1, the saturated speckle clearly exhibits dark structures corresponding to the location of vortices (some are even isolated enough to yield round-shaped dark holes). Fully developed speckle patterns indeed contain a small fraction of non-zero minima (Freund, *Waves in Random Media* 8, 119 (1998)), which means that almost all intensity minima are associated with an optical vortex.

Since we agree that attributing resolution improvement to vortices is partly a matter of interpretation, we slightly modified the related sentence, better arguing our interpretation:

“Here, we attribute the axial resolution improvement to the presence of nodal optical vortex lines dominating the contrast (and thus the cross-correlation product) in the saturated regime, while contrast is dominated by bright “speckle grains” in the linear excitation regime.

To address the question of the role of vortices in the super-resolution process, we drawn Fig. S9 and S10 in the supplementaries. Fig. S9 demonstrates the role of the axial component of the field in the intensity contrast. Yet, the axial field is smaller than the transverse field, except at its vortices. In Fig. S10, we

demonstrate that linear polarization, which minimizes the axial component of the field at vortices strongly elongated along the polarization direction, makes the optical transfer function anisotropic. Circular polarization is actually not used to minimize the *average* axial field at vortices but to minimize the axial field at *isotropic* vortices. We made this clearer in the beginning of the Results section:

“The random wave entering the objective is circularly polarized in order to minimize the axial field at isotropic vortices of same handedness and so, to provide isotropic super-resolution in transverse planes under saturated excitation conditions (see Section S6)”

- In Figure 2i the authors show line profiles of imaged 100nm beads to assess the resolution of their imaging system - with FWHM of 147 nm for the saturated illumination.

In the main text associated with this figure the “effective PSF” is claimed to be 100 nm and 3.4 times below the diffraction limit. What is being used as the metric for the “effective PSF” in this case?

The “effective psf” refers to the rebuilt image that would be obtained if imaging a point like object. Here the beads diameter both for psf characterization and imaging is $d=100\text{nm}$ and cannot be considered as point-like objects. When imaging such beads, the resulting FWHM “ W ” ($=147\text{nm}$) of the rebuilt image is $W=\sqrt{d^2+w^2}$, where “ w ” is the FWHM of the “effective psf”. We thus obtain $w=108\text{nm}$.

A clearer sentence has been added about our estimation of resolution improvement (p.6):

“The size of the bead used for the SPSF characterization limits the utmost achievable resolution, but smaller fluorescent beads yielded too low signal to background ratios. [...] Line profiles plotted in Fig. 2i show a resolution down to 147 nm in the saturated case. The width w of the effective point spread function can be estimated by removing the contribution of the bead size d from the width of the bead image W according to $W = \sqrt{d^2 + w^2}$, so giving $w = 108 \text{ nm}$: a factor 3.3 below the diffraction limit ($0.514\lambda/\text{NA} = 355 \text{ nm}$ for $\text{NA} = 0.77$).”

In the introduction, the increase in resolution is estimated to be 2.3 fold. I could obtain this value by comparing the diffraction limit of their objective to the 147 nm bead line profile FWHM. However, this metric is flawed, since they compare FWHM of the line profile in case of the saturated illumination to the diffraction limit as $\lambda/2\text{NA}$. For fair comparison, it seems that FWHM of the Airy disk should be used ($0.4 \lambda/\text{NA}$) which changes the resolution increase factor to ~ 1.88 .

We agree that $\lambda/2\text{NA}$ is an approximation of the Airy disk FWHM but not so much as the reviewer says. The actual FWHM of an Airy disk is $0.514\lambda/\text{NA}$. This question is closely related to the former one and we think the former comment satisfactorily answer this one also.

It would be beneficial if the authors could in general state more clearly how they obtain all these numbers, i.e. how they actually define their resolution.

We hope the changes detailed above now makes the quantification of resolution improvement clearer.

- Could the authors provide axial cross-sections of the beads in figure 4d (top)? It would be easier to visualize the resolution increase.

As suggested by the reviewer, axial line profiles of the three beads (now in Fig. 2d,e) have now been

plotted in Fig. S11 and exhibit the better resolution improvement for the bead closer to the coverslip. A sentence was also added in the main text to summarize the result of Fig. S11:

“Line profiles of the beads along the axial coordinate (Fig. S.11) exhibit an average axial full width at half maximum $\delta z = 2.5 \pm 0.1\mu\text{m}$ in the non-saturated regime and varies from $\delta z' = 1.7\mu\text{m}$ for the bottom bead closer to the coverslip, to $\sim 2.0\mu\text{m}$ (two other beads) in the saturated regime.”

- Please be careful about typos, in the main text (orthogonal, arsing), as well as in the supplementary material (fluorescence).

We apologize for typos and thank the reviewer for pointing them out. We fixed the problems.

- Some references are listed but not mentioned in the paper itself, mostly from reference 33.

We thank the reviewer for his careful reading, we fixed the problem.

Reviewers' comments:

Reviewer #1 (Remarks to the Author):

The authors have revised their manuscript according to the comments I have indicated in my previous review. They did it well. Although in my opinion the super resolved improvement is still not very significantly visible, I think that the additional experiments in which the authors added images of lysosomes in fixed cultured cells, indeed contributed to the validity of the manuscript. I thus, recommend accepting the revised version of the manuscript for publication.

Reviewer #2 (Remarks to the Author):

I have read the authors response to my comments and found that they have done a nice work in correcting the manuscript.

I think that this manuscript now warrants publication.

Reviewer #3 (Remarks to the Author):

The authors addressed most of my concerns and thoroughly modified the paper, adding some new experimental results to the main text, as well as some more detailed analyses in the supplementary materials.

They now present clearly how the 3D speckle PSF enabled to reconstruct sparse 3D objects using a minimization algorithm fed with sparsity constraints. Moreover, they show that when saturating the fluorescence excitation transition, it can yield super-resolution images.

As opposed to conventional structured illumination methods, this method does not discard the out-of-focus contributions, and is making use of all fluorescence photons, which really matters in a scenario where phototoxicity is an important limitation. Furthermore, these out-of-focus photons do not contribute to the background, as was the case when using plane by plane Wiener deconvolution.

Therefore I would now recommend this paper for publication in Nature communications, under the condition that the authors replace the data in Figure 3 with a more suitable example. I believe their method works, but the figure as it is does a poor job at demonstrating that. The new figures need to (visually) stress out that (1) point scanning requires way more time (and therefore total energy on sample), (2) Wiener deconvolution is not performing as well as the new approach (the threshold makes it artificially look similar).

I understand that the authors want to point out the fact that the transverse super-resolution is only provided by the saturated speckle illumination, but this information is already provided in Fig.2.

The volumetric visualization in Fig.3d is not extremely informative, and in my opinion should be replaced by cross-sections. The illustration of the following sentence is not so satisfying to me: "Noteworthy axial resolution is better with FISTA since less prone to yield reconstruction artifacts from out-of-focus planes, but transverse resolutions are similar in both reconstruction schemes, meaning that super-resolution is only obtained thanks to optical saturation."

I note that the use of biologically relevant samples is a praiseworthy effort, but overall the scanned region (as indicated in Fig.3b) does not exhibit features that nicely demonstrate the capabilities of the technique, at least at the first glimpse. And this is rather unfortunate for a paper demonstrating a new imaging technique.

Although the authors performed a large amount of work, there are still some glitches.

* abstract, l.6: fading/fatigue

* discussion, l.5: "Here, samples were imaged thanks to 2D raster scans with but, in principle:"

* The following sentence (response to reviewer 2) is not in the introduction: "The vectorial nature of light cannot be neglected in tightly focused beams, especially under saturated conditions, in which case the axial field may destroy most of intensity zeros."

In general it would have been convenient to have line numbers of the added sentences.

* page 2, paragraph 2 of Results section, line 4: "A sub-nanosecond laser is used, delivering pulses shorter than the fluorescence life-time of the dye ($\tau_f \sim 500$ ps)"

* Captions:

- Fig.1: what are the dotted lines in h and I?

- Fig.2: please add more details to the description of m, even if straightforward there is no proper explanation of what we see.

- Fig.3: there is no scale bar in d.

More generally, I would advise the authors to write the figures captions in such a way that every detail is explained.

We thank the reviewers for advising publication of our manuscript. We addressed the remaining remarks and comments still raised by reviewers #1 and #3. In particular, we significantly modified Fig. 3 to better highlight the performances of our new imaging modality. Details are appended below.

Reviewer #1 (Remarks to the Author):

The authors have revised their manuscript according to the comments I have indicated in my previous review. They did it well. Although in my opinion the super resolved improvement is still not very significantly visible, I think that the additional experiments in which the authors added images of lysosomes in fixed cultured cells, indeed contributed to the validity of the manuscript. I thus, recommend accepting the revised version of the manuscript for publication.

We thank the reviewer for advising publication. Also in agreement to reviewer#3's requests, we modified Fig. 3 in the manuscript in order to better highlight:

- a- Resolution improvement under saturated conditions
- b- Reconstruction improvement by FISTA
- c- the 3D imaging capability

We agree that the former Fig.3 did not highlight resolution improvement well enough. In particular, we now enlarged images without linear interpolation so that pixels are visible making resolution improvement more visible.

Reviewer #2 (Remarks to the Author):

I have read the authors response to my comments and found that they have done a nice work in correcting the manuscript.

I think that this manuscript now warrants publication.

We thank the reviewer for their congratulations and for advising publication.

Reviewer #3 (Remarks to the Author):

The authors addressed most of my concerns and thoroughly modified the paper, adding some new experimental results to the main text, as well as some more detailed analyses in the supplementary materials.

They now present clearly how the 3D speckle PSF enabled to reconstruct sparse 3D objects using a minimization algorithm fed with sparsity constraints. Moreover, they show that when saturating the fluorescence excitation transition, it can yield super-resolution images.

As opposed to conventional structured illumination methods, this method does not discard the out-of-focus contributions, and is making use of all fluorescence photons, which really matters in a scenario

where phototoxicity is an important limitation. Furthermore, these out-of-focus photons do not contribute to the background, as was the case when using plane by plane Wiener deconvolution.

We thank the reviewer for their careful reading of our manuscript and for pointing out the novelty of our technique and the important interest for reducing phototoxicity.

Therefore I would now recommend this paper for publication in Nature communications, under the condition that the authors replace the data in Figure 3 with a more suitable example. I believe their methods works, but the figure as it is does a poor job at demonstrating that.

We thank the reviewer for pointing out the lack of readability of our Fig.3. We agree the former version of the figure was actually too cumbered, making it hard to identify regions of interest in particular. We have now significantly modified Fig. 3 in order to improve visual evidences. The new figure now better illustrates:

- a- Resolution improvement under saturated conditions
by selecting and zooming on proper regions of interest.
- b- Reconstruction improvement by FISTA
both in terms of signal to noise (threshold removed in Wiener-deconvolved data) and out-of-focus signal sectioning (thanks to the reviewers' suggestion to draw cross-sections rather than 3D views)
- c- the 3D imaging capability
thanks to the cross-sections shown in new Fig. 3f suggested by the reviewer.

The new figures needs to (visually) stress out that (1) point scanning requires way more time (and therefore total energy on sample),

We thank the reviewer for suggesting further stressing upon this advantage of our technique. We fully agree that such visual evidence is highly valuable to the manuscript. We thus added in Fig. 3a two schemes to recall that 3D scanning is required to get a point scanning image while a single 2D speckle image allows getting the full 3D image with compressive microscopy.

(2) Wiener deconvolution is not performing as well as the new approach (the threshold makes it artificially look similar).

I understand that the authors want to point out the fact that the transverse super-resolution is only provided by the saturated speckle illumination, but this information is already provided in Fig.2.

We agree with the reviewer that it is better to remove the threshold applied to the Wiener deconvolved data. We thus removed the threshold, which now allows a clearer and fairer comparison with FISTA results.

The volumetric visualization in Fig.3d is not extremely informative, and in my opinion should be replaced by cross-sections. The illustration of the following sentence is not so satisfying to me:

“Noteworthy axial resolution is better with FISTA since less prone to yield reconstruction artifacts from out-of-focus planes, but transverse resolutions are similar in both reconstruction schemes, meaning that super-resolution is only obtained thanks to optical saturation.”

We thank the reviewer for their suggestion: we replaced the 3D profiles by cross-sections and we agree that it better demonstrates axial resolution improvement while allowing visualizing the signal to noise ratio (which could not be so well appreciated with 3D views). The threshold removal for Wiener deconvolved data in the former version of figure 3 also contributed to this lack of evidence that out-of-focus objects contribute to noise.

I note that the use of biologically relevant samples is a praiseworthy effort, but overall the scanned region (as indicated in Fig.3b) does not exhibit features that nicely demonstrate the capabilities of the technique, at least at the first glimpse. And this is rather unfortunate for a paper demonstrating a new imaging technique.

We acknowledge that the former version of the figure did not allow fully appreciating the performances of our approach. Thanks to the reviewer’s criticisms and suggestions, we now feel our new images of Lysosomes are perfectly demonstrative.

Although the authors performed a large amount of work, there are still some glitches.

* abstract, l.6: fading/fatigue

We understand that this writing was confusing. We removed “fatigue”.

* discussion, l.5: "Here, samples were imaged thanks to 2D raster scans with but, in principle:"

Fixed.

* The following sentence (response to reviewer 2) is not in the introduction: “The vectorial nature of light cannot be neglected in tightly focused beams, especially under saturated conditions, in which case the axial field may destroy most of intensity zeros.”

We apologize for this mistake. This sentence was accidentally removed in rephrasing operations. We added it again so that the paragraph now reads:

“Optical vortices in speckles are associated with nodal lines of intensity (in 3D) that can confine fluorescence to sub-diffraction dimensions [29], despite the contribution of the axial field. Indeed, in tightly focused beams, the vectorial nature of light cannot be neglected, especially under saturated conditions, in which case the axial field may destroy most of intensity zeros.”

In general it would have been convenient to have line numbers of the added sentences.

* page 2, paragraph 2 of Results section, line 4: “A sub-nanosecond laser is used, delivering pulses shorter than the fluorescence life-time of the dye ($\tau_p \sim 500$ ps)”

Fixed.

* Captions:

- Fig.1: what are the dotted lines in h and l?

Dotted circles are drawn as a visual reference for easier comparison between spectra boundaries in the linear and in the saturated excitation regime. We added this missing information to the caption:

“(dotted circles in h and i materialize the spectrum boundary in i, as a visual reference)”

- Fig.2: please add more details to the description of m, even if straightforward there is no proper explanation of what we see.

The caption was completed with the following sentence:

“Axial resolution improvement is shown in m where an axial bead intensity profile is plotted both in the linear and in the saturated excitation regimes”

- Fig.3: there is no scale bar in d.

Scale bar was added.

More generally, I would advise the authors to write the figures captions in such a way that every detail is explained.

We fully agree with the reviewer and tried to achieve so. We thank them for pointing out missing information.

REVIEWERS' COMMENTS:

Reviewer #3 (Remarks to the Author):

All comments were addressed. I support publication of this manuscript.